# Frequency and Characteristics of Non-Neurological and Neurological Stroke Mimics in the Emergency Department

**DOI:** 10.3390/jcm12227067

**Published:** 2023-11-13

**Authors:** Jordi Kühne Escolà, Bessime Bozkurt, Bastian Brune, Woon Hyung Chae, Lennart Steffen Milles, Doreen Pommeranz, Lena Brune, Philipp Dammann, Ulrich Sure, Cornelius Deuschl, Michael Forsting, Clemens Kill, Christoph Kleinschnitz, Martin Köhrmann, Benedikt Frank

**Affiliations:** 1Department of Neurology and Center for Translational Neuro- and Behavioral Sciences (C-TNBS), University Hospital Essen, 45147 Essen, Germany; jordi.kuehneescola@uk-essen.de (J.K.E.); bessime.bozkurt@uk-essen.de (B.B.); woonhyung.chae@uk-essen.de (W.H.C.); lennart.milles@uk-essen.de (L.S.M.); doreen.pommeranz@uk-essen.de (D.P.); lenabrune@gmx.de (L.B.); christoph.kleinschnitz@uk-essen.de (C.K.); martin.koehrmann@uk-essen.de (M.K.); 2Department of Trauma, Hand and Reconstructive Surgery, University Hospital Essen, 45147 Essen, Germany; bastian.brune@uk-essen.de; 3Medical Emergency Service of the City of Essen, 45139 Essen, Germany; 4Department of Neurosurgery and Spine Surgery, University Hospital Essen, 45147 Essen, Germany; philipp.dammann@uk-essen.de (P.D.); ulrich.sure@uk-essen.de (U.S.); 5Institute of Diagnostic and Interventional Radiology and Neuroradiology, University Hospital Essen, 45147 Essen, Germanymichael.forsting@uk-essen.de (M.F.); 6Center of Emergency Medicine, University Hospital Essen, 45147 Essen, Germany; clemens.kill@uk-essen.de

**Keywords:** stroke mimics, suspected stroke, stroke diagnosis, prehospital, non-neurological

## Abstract

Background: Stroke mimics are common in the emergency department (ED) and early detection is important to initiate appropriate treatment and withhold unnecessary procedures. We aimed to compare the frequency, clinical characteristics and predictors of non-neurological and neurological stroke mimics transferred to our ED for suspected stroke. Methods: This was a cross-sectional study of consecutive patients with suspected stroke transported to the ED of the University Hospital Essen between January 2017 and December 2021 by the city’s Emergency Medical Service. We investigated patient characteristics, preclinical data, symptoms and final diagnoses in patients with non-neurological and neurological stroke mimics. Multinominal logistic regression analysis was performed to assess predictors of both etiologic groups. Results: Of 2167 patients with suspected stroke, 762 (35.2%) were diagnosed with a stroke mimic. Etiology was non-neurological in 369 (48.4%) and neurological in 393 (51.6%) cases. The most common diagnoses were seizures (23.2%) and infections (14.7%). Patients with non-neurological mimics were older (78.0 vs. 72.0 y, *p* < 0.001) and more likely to have chronic kidney disease (17.3% vs. 9.2%, *p* < 0.001) or heart failure (12.5% vs. 7.1%, *p* = 0.014). Prevalence of malignancy (8.7% vs. 13.7%, *p* = 0.031) and focal symptoms (38.8 vs. 57.3%, *p* < 0.001) was lower in this group. More than two-fifths required hospitalization (39.3 vs. 47.1%, *p* = 0.034). Adjusted multinominal logistic regression revealed chronic kidney and liver disease as independent positive predictors of stroke mimics regardless of etiology, while atrial fibrillation and hypertension were negative predictors in both groups. Prehospital vital signs were independently associated with non-neurological stroke mimics only, while age was exclusively associated with neurological mimics. Conclusions: Up to half of stroke mimics in the neurological ED are of non-neurological origin. Preclinical identification is challenging and a high proportion requires hospitalization. Awareness of underlying etiologies and differences in clinical characteristics is important to provide optimal care.

## 1. Introduction

Stroke mimics are common in the neurological emergency department (ED). Between 20% and up to more than 40% of patients evaluated for suspected stroke are diagnosed with a non-stroke disease [1,2,3]. Underlying etiologies are heterogeneous and can include serious emergencies of non-neurological or neurological origin that require distinct approaches towards diagnosis and treatment [4,5,6]. Delayed identification or misdiagnosis can expose patients to unnecessary procedures, engender costs to the healthcare system and defer initiation of an appropriate treatment for the underlying disease [7]. Thus, early recognition of stroke mimics is important to optimize patient care and utilization of limited resources. Nonetheless, differentiating stroke from non-stroke conditions can be challenging, especially in the preclinical setting where diagnostic resources and neurological expertise are limited.

Previous analyses stratified non-neurological and neurological stroke mimic etiologies and found that up to 44.9% of stroke mimics in patients transferred to a comprehensive stroke center for suspected stroke by the Emergency Medical Service (EMS) were of non-neurological origin [8].

Demographic characteristics and clinical features such as younger age, female sex, transient or unspecific symptoms and level of consciousness alteration have found to be associated with misdiagnosis in patients with suspected stroke and it has been hypothesized that these features and thus reasons for misinterpretation may differ between subgroups of stroke mimics [9]. Understanding these differences could help clinicians to better recognize different stroke mimic etiologies and facilitate decision-making regarding proper diagnostics and treatment. However, these aspects have not been investigated further and the data on non-neurological stroke mimics are still sparse. We therefore investigated prehospital symptom patterns and clinical characteristics in patients with non-neurological and neurological stroke mimics who were transferred to our neurological ED by the EMS as suspected stroke patients. The objective of this study was to answer three questions: (1) What is the proportion of non-neurological and neurological stroke mimics among patients transferred by the EMS to an urban comprehensive stroke center for suspected stroke? (2) Do preclinical characteristics and symptoms differ in patients with non-neurological and neurological stroke mimics? (3) Which variables lead to misdiagnosis by the EMS in stroke mimics of non-neurological or neurological etiology?

## 2. Materials and Methods

### 2.1. Study Design and Data Source

This was a retrospective, cross-sectional study of consecutive patients transported to the neurological ED of the University Hospital Essen between January 2017 and December 2021 by the city’s EMS. All cases of suspected stroke by the EMS personnel in adults (≥18 years) who were then evaluated at our neurological ED were included. Excluded cases were those without neurological examination and patients who were discharged against medical advice before a final diagnosis could be established. Since our goal was to focus on preclinical characteristics in our analysis, this study also excluded patients who were evaluated for suspected in-house stroke, referrals from other hospitals and other cities and patients who presented themselves on their own initiative.

Final diagnoses at discharge as made by the treating physician were classified as cerebrovascular events (ischemic stroke, transient ischemic attack, intracerebral hemorrhage; CVE) or stroke mimics. For this analysis, we primarily focused on patients with stroke mimics. These were independently reviewed by two stroke neurologists (J.K.E., B.F.), who further stratified the etiology as non-neurological or neurological. Final categorization was then compared and in cases of differing opinions, cases were discussed between the two until a consensus was achieved. Since our goal was to compare non-neurological and neurological stroke mimics, we excluded *n* = 9 patients in which no specific diagnosis could be made despite a complete workup and therefore could not be assigned to either of the groups.

Preclinical data including systolic and diastolic blood pressure, heart rate, blood glucose and time metrics were extracted from the EMS records. Information on demographics, medical history, symptoms as well as further diagnostic procedures and treatment was retrieved from the hospital’s clinical documentation, ED records and final discharge letters. To further investigate symptom patterns that triggered emergency personnel suspicion of stroke in patients who were ultimately diagnosed with a non-stroke disease, we reviewed each case for the primary clinical manifestation and retrospectively determined the leading symptom. We also recorded whether symptoms were of a new onset or due to residual deficits of a known stroke or other prior neurological disease.

This study was approved by the ethics committee of the medical faculty of the University Duisburg-Essen (approval number 18-8408-BO) and conducted under the standards of the local data protection authority. It was performed in accordance with the principles of the Declaration of Helsinki. Requirement for written consent was waived by the ethics committee.

### 2.2. Statistical Analyses

We used descriptive statistics to compare final diagnoses, preclinical and baseline data, symptoms upon admission, as well as the intrahospital management of non-neurological and neurological stroke mimics. For the categorical data, we report counts and percentages, while continuous data are described as the mean (standard deviation) or median (interquartile range). Chi-square or Mann–Whitney U tests were used for further comparison between non-neurological or neurological mimics, as appropriate.

Multinomial logistic regression analysis was used to explore the association between patient-related, EMS-ascertainable predictors and diagnosis of a non-neurological or neurological stoke mimic. Baseline and preclinical variables were selected based on significance on univariate multinominal regression analysis and included in a multivariate model adjusted for age, sex and other comorbidities. We excluded non-patient-related organizational parameters such as time metrics.

All statistical tests were two sided, and *p*-values of <0.05 were considered statistically significant. No adjustment was made for multiple testing. Statistical analyses were performed using SPSS, version 29 (IBM Corp., Armonk, NY, USA).

## 3. Results

Between January 2017 and December 2021, 2242 patients were transported to the neurological ED of the University Hospital Essen via EMS for suspected stroke (Figure 1). Of the 2167 patients eligible for further analysis, 1405 (64.8%) received a diagnosis of ischemic stroke, TIA or intracranial hemorrhage, while 762 (35.2%) were diagnosed with a stroke mimic. The etiology was regarded non-neurological in 369 (48.4%) and neurological in 393 (51.6%) cases. The most common non-neurological diagnoses were infections (14.7%), cardiovascular disorders (11.3%) and exsiccosis (8.1%). Seizures (23.2%), peripheral neuropathy (5.6%) and migraine (4.6%) were the most frequent neurological mimics (Figure 2).

Compared to patients with cerebrovascular events, those who had a stroke mimic were more likely to have chronic kidney or liver disease and had a lower prevalence of atrial fibrillation and hypertension (Table 1). Prehospital systolic and diastolic blood pressure was lower, Blood glucose levels and heart rate were higher and EMS time metrics were longer in patients with stroke mimics. More than half of patients were women, with no difference between either of the groups. Patients with non-neurological mimics were older (78.0 years, IQR 63.0–85.5 vs. 72.0 years, IQR 59.0–82.0; *p* < 0.001), had a higher prevalence of chronic kidney disease (17.3% vs. 9.2%, *p* < 0.001) and heart failure (12.5% vs. 7.1%, *p* = 0.014) as well as a lower prevalence of malignancy (8.7% vs. 13.7%, *p* = 0.031) as compared to patients who were diagnosed with a neurological mimic. Other vascular risk factors and comorbidities such as atrial fibrillation, coronary artery disease, diabetes, hypertension, peripheral artery disease or a history of stroke were balanced between groups. More than half of patients had at least one of the abovementioned vascular risk factors and vascular comorbidities (58.8% vs. 56.0%, *p* = 0.464).

### 3.1. Prehospital Symptom Patterns

When assessing symptoms that triggered prehospital misdiagnosis of stroke, we found that patients with non-neurological stroke mimics more often presented with unexplained quantitative or qualitative level of consciousness alterations (32.8% vs. 24.2%; *p* = 0.008 and 21.2% vs. 14.5%; *p* = 0.018, respectively) and dysarthria (11.4% vs. 3.8%, *p* < 0.001) as the leading clinical manifestation (Figure 3). They were less likely to present due to facial droop (1.6 vs. 8.7%; *p* < 0.001), limb ataxia (2.4% vs. 8.1%, *p* < 0.001), limb paresis (9.5% vs. 15.8%, *p* = 0.010) or gaze deviation (none vs. 1.5%, *p* = 0.031). In the majority of cases (85.4%), preclinical misdiagnosis of stroke was based on new-onset symptoms ultimately caused by non-neurological or neurological non-stroke diseases. In contrast, residual symptoms of a known stroke or other neurological diseases triggered prehospital misdiagnosis of stroke in only 14.6% (11.4% vs. 17.6%, *p* < 0.018) of cases.

### 3.2. Clinical Characteristics upon Admission

Upon arrival at our neurological ED, overall National Institutes of Health Stroke Scale (NIHSS) scores were lower in patients with non-neurological stroke mimics (median NIHSS 1, IQR 0–3 vs. 2, IQR 0–4; *p* < 0.001) and focal neurological symptoms were less common (any focal symptom 38.8 vs. 57.3%, *p* < 0.001). The largest differences were observed for facial droop (8.7 vs. 20.9%; *p* < 0.001) and limb paresis (18.4% vs. 28.2%; *p* = 0.002) (Figure 4). However, aphasia (10.3% vs. 15.0%, *p* = 0.064) and sensory impairment (3.4% vs. 6.6%, *p* = 0.069) showed trends in the same direction. The level of consciousness impairment was more common in non-neurological mimics (14.9% vs. 7.9%, *p* = 0.003).

### 3.3. Intrahospital Management

Over two-thirds of patients with stroke mimics received diagnostic brain imaging (Table 2). Magnetic resonance imaging (MRI) was less frequent in patients with a non-neurological etiology (3.3% vs. 13.2%, *p* < 0.001) and there were similar trends for non-contrast CT (66.4% vs. 72.5%, *p* = 0.070). There were no differences in other imaging modalities including CT angiography (17.6% vs. 20.4%, *p* = 0.357) and CT perfusion (9.2% vs. 10.9%, *p* = 0.471). A total of 7/762 patients (<1%) were treated with intravenous thrombolysis for suspected ischemic stroke with no difference between groups (*n* = 4, 1.1% vs. *n* = 3, 0.8%; *p* = 0.718). No bleeding complications occurred in thrombolyzed patients. Hospital admission was less common in patients with non-neurological mimics (39.3% vs. 47.1%, *p* = 0.034); however, hospitalization lasted longer as compared to patients with a neurological etiology (median length of hospitalization 6 days, IQR 3–12 vs. 4 days, IQR 2–7; *p* = 0.014).

### 3.4. Predictors of Non-Neurological and Neurological Stroke Mimics

After adjusting for age, sex and other comorbidities, chronic kidney disease (OR 8.744, 95% CI 5.010–15.261 and 4.994, 2.770–9.004) and liver disease (OR 3.011, 95% CI 1.414–6.412 and 2.232, 1.007–4.945) remained positive predictors of being diagnosed with a stroke mimic in patients with non-neurological as well as neurological etiologies (Appendix A, Table A1). Atrial fibrillation (OR 0.377, 95% CI 0.258–0.551 and 0.540, 0.393–0.772) and hypertension (OR 0.595, 95% CI 0.431–0.82 and 0.573, 0.419–0.783) were independent negative predictors of both stroke mimic groups. Prehospital vital signs remained predictors of being diagnosed with a non-neurological stroke mimic only (increasing systolic blood pressure OR 0.988, 95% CI 0.982–0.994; increasing diastolic blood pressure OR 0.989, 95% CI 0.979–0.999; increasing heart rate OR 1.008, 95% CI 1.000–1.016; increasing blood glucose levels OR 1.003, 95% CI 1.000–1.005). Heart failure (OR 2.178, 95% CI 1.286–3.691), diabetes (OR 0.650, 95% CI 0.449–0.939) and malignancy (OR 0.514, 95% CI 0.314 –0.842) were also exclusively associated with a non-neurological stroke mimic diagnosis, while increasing age was an independent negative predictor of neurological stroke mimics only (OR 0.982, 95% CI 0.972–0.992).

## 4. Discussion

One third of patients transferred to our comprehensive stroke center by EMS for suspected stroke were finally diagnosed with a stroke mimic and nearly half of the underlying pathologies were of non-neurological origin. More than 50% of stroke mimic patients had vascular risk factors and/or focal–neurological symptoms and two-fifths required hospital admission.

Regarding the rate and diagnostic spectrum of stroke mimics as well as differences in baseline characteristics between non-neurological and neurological etiologies, our findings are well comparable to previous studies [8,10], as patients with non-neurological mimics were older and had more comorbidities. Several authors have also assessed predictors of being diagnosed with a stroke mimic, among which absence of vascular risk factors, a younger age and female sex have been highlighted in particular [2,11,12]. While our study confirmed many of these previous observations, it provides closer insight towards differences in these predictors among distinct stroke mimic etiologies. While chronic kidney and liver disease (positive) as well as atrial fibrillation and hypertension (negative) were independent predictors of being diagnosed with a stroke mimic regardless of its etiology, other comorbidities (heart failure diabetes, malignancy) as well as prehospital vital signs were exclusively associated with non-neurological stroke mimics. Increasing age, in turn, was an independent negative predictor of neurological stroke mimics only. While these findings could help clinicians to identify different types of stroke mimics in certain cases, this more differential perspective on stroke mimic predictors highlights that their application on an individual patient level warrants caution. In our study, some previously described predictors were only present in a certain subgroup of stroke mimics (age, prehospital blood pressure), while others were not at all associated with a stroke mimic diagnosis (female sex). These important barriers might become more important in the future, as healthcare providers are likely to face higher numbers of older and more comorbid patients with stroke mimics due to demographic changes [13].

Examination of prehospital symptom patterns revealed that the majority of patients with non-neurological stroke mimics are misdiagnosed as stroke due to impaired consciousness. With a wide range of underlying etiologies reaching from benign and transient conditions to life-threatening events, the level of consciousness alterations can pose substantial challenges for healthcare providers. In patients with suspected stroke, they have been found to independently predict being diagnosed with a stroke mimic [14]. Additional studies may target these presentations in order to refine prehospital triage.

Nonetheless, the overall presence of focal neurological symptoms was high, even in patients with non-neurological stroke mimics. Most of these manifestations were not explained by residual symptoms of an old stroke or another neurological disease. However, the fact that primarily non-neurological conditions can manifest with focal neurological symptoms has been described previously, particularly in speech disorders [15,16], and our study is concordant with these observations. Although the underlying mechanisms are not fully understood, concomitant encephalopathy is likely to be a contributing factor. Several non-neurological stroke mimic etiologies such as infections, metabolic/toxic conditions or hypertensive crisis can be associated with encephalopathy [17,18,19,20], which in turn can cause mental status alterations but also focal neurological symptoms such as speech disorders or limb paresis. Clinicians should bear this in mind when investigating patients with suspected stroke and exclude non-neurological causes in uncertain cases.

Various tools have been developed to improve diagnostic accuracy in patients with suspected stroke and guide decision-making in the acute phase, including scores for the detection of stroke mimics in the emergency department and via telestroke medicine [21,22,23]. Their application in the prehospital setting has been investigated but lacks prospective validation [24]. A recent study also highlighted the potential of prehospital telestroke assessment, which was found superior in predicting candidates for reperfusion therapies when compared to a standardized in-person EMS triage [25]. Although the application of scores and use of telemedicine may refine triage protocols and bear the potential of improving prehospital diagnostic performance, our findings highlight an important caveat. Identification of stroke mimics is challenging, as etiologies and clinical presentations are heterogeneous and include transient events, altered mental status or focal symptoms. In patients with suspected stroke, however, a high sensitivity is considered the most important aspect when making a diagnosis, as the harm of missing a stroke outweighs the burden of preclinical overdiagnosis (false-positive suspected strokes). Thus, thresholds for suspecting a stroke and transferring patients to a corresponding stroke center should remain low. The final diagnosis is up to the treating physicians of the stroke center. Furthermore, in our study more than a third of patients with non-neurological stroke mimics and nearly half with a neurological non-stroke etiology still had to be admitted to our hospital. These findings indicate that even beyond diagnosis and treatment of cerebrovascular events, many patients with suspected stroke still require further neurological expertise or multidisciplinary approaches. Thus, transfer to comprehensive care centers is justified.

Several limitations should be considered when interpreting our study. Due to the retrospective design, findings should be regarded as hypothesis-generating and non-confirmatory and, especially, our findings on potential predictors must be interpreted with caution. All data were collected in a single tertiary center, which limited their generalizability. Focusing on EMS-suspected stroke from a single city and excluding patients who underwent secondary transportation to our comprehensive stroke center from other hospitals or other cities as well as in-hospital strokes bore the risk of selection bias; however, as a comprehensive stroke center, most referrals from other hospital and cities of patients with “suspected stroke” were for acute treatment of actually confirmed stroke and thus, including these patients would have underestimated the rate of stroke mimics. We therefore believe that our cohort reasonably represents the spectrum of suspected stroke in urban Germany. Despite screening a large number of consultations for suspected stroke, the sample size of stroke mimic subgroups remained relatively small and there was an important proportion of missing prehospital data. We were also not able to calculate scores for the detection of stroke mimics in our cohort because of missing information on their history of prior seizures. Using the National Institutes of Health Stroke Scale to characterize symptom presentations bore the risk of oversimplification, as the scores were heavily weighted on anterior circulation stroke and did not include other important clinical features of suspected stroke such as imbalance, vertigo, fine motor impairment or other non-focal symptoms. In addition, patients in our emergency department might have been misdiagnosed, leading to potential over- or underestimation of the stroke mimic rate and overestimation of focal symptoms in supposedly non-neurological stroke mimics who had an actual stroke.

Nonetheless, we provide real-world data from a large cohort of consecutive patients with suspected stroke. To our knowledge, this is the first study to combine a detailed stratification of non-neurological and neurological stroke mimic etiologies with an analysis of symptom patterns, clinical characteristics and predictors of stroke mimics in both groups.

## 5. Conclusions

Up to half of stroke mimics transported to the neurological ED by the EMS are of non-neurological origin. Preclinical identification is challenging and misdiagnosis often occurs in patients presenting with an impaired level of consciousness. Additionally, high age and the presence of vascular risk factors and focal symptoms pose substantial difficulties. Preclinical predictors might be useful for early identification of different stroke mimic etiologies; however, prospective studies are required and thresholds for suspecting stroke and initiating transfer to a corresponding stroke center should remain low in the preclinical setting, as even an important proportion of patients with stroke mimics requires hospital admission. Nonetheless, clinicians should be aware of differences in clinical presentations of non-neurological and neurological stroke mimics to engage the right approach towards diagnosis and treatment.

## Figures and Tables

**Figure 1 jcm-12-07067-f001:**
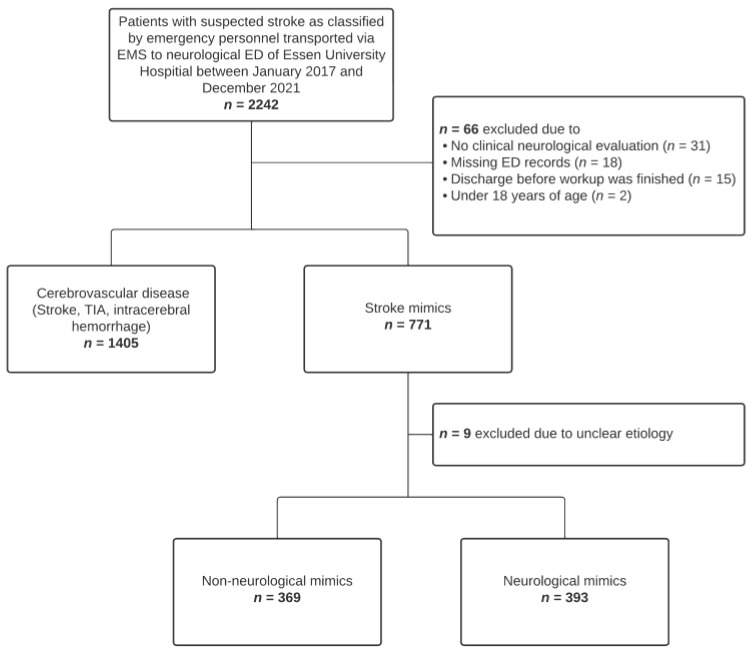
Flow chart of the study population. Abbreviations: ED, emergency department; EMS, emergency medical service; TIA, transient ischemic attack.

**Figure 2 jcm-12-07067-f002:**
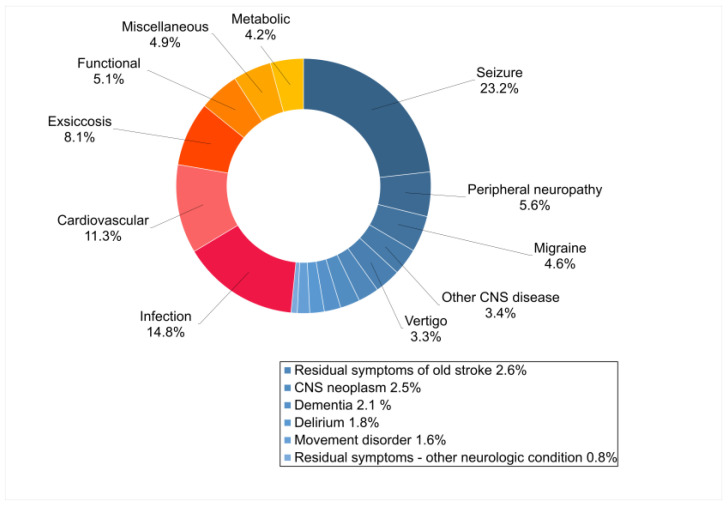
Final diagnoses in *n* = 762 patients with non-neurological and neurological stroke mimics. Abbreviations: CNS, central nervous system. Percentages do not round up to exactly 100% due to rounding.

**Figure 3 jcm-12-07067-f003:**
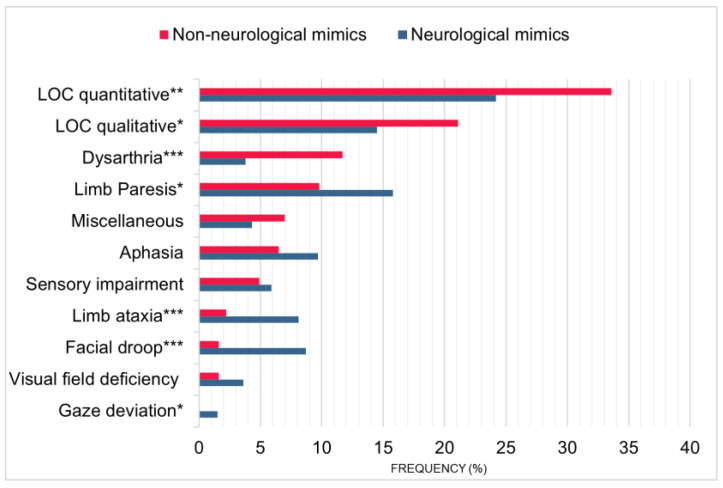
Frequency (%) of leading symptoms in *n* = 762 patients with non-neurological and neurological stroke mimics. The asterisks indicate a *p* < 0.05 (*), *p* < 0.01 (**) and *p* < 0.001 (***). Abbreviations: LOC, level of consciousness.

**Figure 4 jcm-12-07067-f004:**
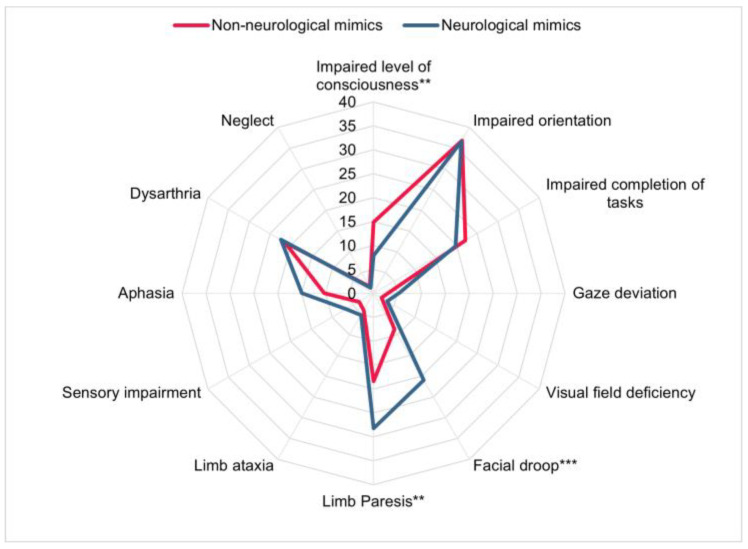
Frequency (%) of National Institutes of Health Stroke Scale (NIHSS) symptoms in *n* = 762 patients with non-neurological and neurological stroke mimics. The asterisks indicate a *p* < 0.01 (**) and *p* < 0.001 (***).

**Table 1 jcm-12-07067-t001:** Baseline information and preclinical data for *n* = 2167 patients with suspected stroke, comparing all patients with cerebrovascular events vs. stroke mimics ^†^ as well as non-neurological vs. neurological stroke mimics ^‡^.

	CVE, All (*n* = 1405)	Mimic, All (*n* = 762)	*p*-Value ^†^	Non-Neurological Mimic (*n* = 369)	Neurological Mimic (*n* = 393)	*p*-Value ^‡^
Age (years)	76 (65.0–84.0)	76.0 (61.0–84.0)	0.025	78.0 (63.0–85.5)	72.0 (59.0–82.0)	<0.001
Female sex	744 (53.0)	417 (54.7)	0.443	202 (54.7)	215 (54.7)	>0.99
Comorbidities and vascular risk factors						
Atrial fibrillation	425 (30.2)	145 (19.0)	<0.001	68 (18.7)	77 (20.3)	0.781
Chronic kidney disease	42 (3.0)	100 (13.1)	<0.001	64 (17.3)	36 (9.2)	<0.001
Coronary artery disease	202 (18.4)	114 (15.0)	0.052	60 (16.3)	54 (13.7)	0.361
Deep vein thrombosis	28 (2.0)	13 (1.7)	0.742	7 (1.9)	6 (1.5)	0.785
Diabetes	428 (30.5)	191 (25.1)	0.008	99 (26.8)	92 (23.4)	0.279
Heart failure	87 (7.9)	74 (9.7)	0.181	46 (12.5)	28 (7.1)	0.014
Hypertension	1071 (76.2)	439 (57.6)	<0.001	216 (58.5)	223 (56.7)	0.660
Liver disease	26 (1.9)	37 (4.9)	<0.001	22 (6.0)	15 (3.8)	0.181
Malignancy	191 (13.6)	87 (11.4)	0.158	33 (8.9)	54 (13.7)	0.040
Peripheral artery disease	81 (5.8)	39 (5.1)	0.557	22 (6.0)	17 (4.3)	0.327
Prior stroke	362 (25.8)	221 (29.0)	0.105	99 (26.9)	122 (31.0)	0.231
Preclinical data						
Time from alarm to arrival at scene (min) ^a^	5.1 (3.7–7.0)	6.5 (4.8–8.5)	<0.001	6.5 (4.8–8.6)	6.5 (4.8–8.4)	0.813
Scene time (min) ^b^	17.3 (12.9–21.9)	18.3 (14.0–23.4)	<0.001	19.3 (15.2–24.5)	17.2 (13.0–22.1)	<0.001
Systolic blood pressure ^c^	158 (139–178)	146 (127–166)	<0.001	140 (120–164)	150 (131–167)	<0.001
Diastolic blood pressure ^d^	90 (78–103)	85 (72–98)	<0.001	81 (68–96)	87 (76– 99)	<0.001
Heart rate (per min) ^e^	75 (48–90)	84 (71–96)	<0.001	86 (71–98)	82 (71–95)	0.121
Blood glucose level (mg/dL) ^f^	120 (81–148)	130 (111–165)	<0.001	136 (113–172)	127 (108–157)	0.004

Data represent the *n* (%) or median (IQR). Abbreviations: CVE, cerebrovascular event (ischemic stroke, transient ischemic attack, intracerebral hemorrhage); mg/dL, milligram per deciliter; min, minutes. Data are available for the subsets as ^a^ *n* = 1345/1405 and 720/762 (342/369 and 378/393), ^b^ *n* = 1279/1405 and 680/762 (327/369 and 353/393), ^c^ *n* = 1036/1405 and 738/762 (358/369 and 380/393), ^d^ *n* = 1035/1405 and 737/762 (358/369 and 379/393), ^e^ *n* = 1364/1405 and 742/762 (360/369 and 382/393), and ^f^ *n* = 1132/1405 and 615/762 (306/369 and 309/393). In non-neurological stroke mimics, prehospital systolic and diastolic blood pressure were lower (140 mmHg, IQR 120–164 vs. 150 mmHg, IQR 131–167; *p* < 0.001 and 81 mmHg, IQR 68–96 vs. 87 mmHg, IQR 76–99; *p* < 0.001, respectively). Prehospital blood glucose levels were higher (136 mg/dL, IQR 113–172 vs. 127 mg/dL, IQR 108–157; *p* = 0.004) and paramedics spent more time at the scene of the emergency (19.3 min, IQR 15.2–24.5 vs. 17.2 min, IQR 13.0–22.1; *p* < 0.001).

**Table 2 jcm-12-07067-t002:** In-hospital data and management of *n* = 762 non neurological vs. neurological stroke mimics.

	Non-Neurological Mimic (*n* = 369)	Neurological Mimic (*n* = 393)	*p*-Value
In-hospital data			
NIHSS at admission	1–0 (0.0–3.0)	2–0 (0.0–4.0)	<0.001
Time from symptom onset to admission (hours) ^a^	4–3 (1.0–13.1)	4–5 (1.1–13.3)	0.839
Intravenous thrombolysis	4 (1.1)	3 (0.8)	0.720
Admission to hospital	145 (39.3)	185 (47.1)	0.034
Length of admission (days) ^b^	6 (2.0–12.0)	4 (2.0–7.0)	0.014
Diagnostic brain imaging			
NCCT	245 (66.4)	285 (72.5)	0.070
CT angiography	65 (17.6)	80 (20.4)	0.357
CT perfusion	34 (9.2)	43 (10.9)	0.471
Post-contrast CT	4 (1.1)	11 (2.8)	0.118
MRI	12 (3.3)	56 (13.2)	<0.001

Data represent the *n* (%) or median (IQR). Abbreviations: CT, computed tomography; MRI, magnetic resonance imaging; NCCT, non-contrast computed tomography; NIHSS, National Institutes of Health Stroke Scale. Data are available for the subsets as ^a^ *n* = 346/369 and 340/393, ^b^ *n* = 140/145 and 183/185. As patients could receive no imaging or different imaging modalities, the total percentages do not add up to 100%.

## Data Availability

Data are available upon reasonable request from the corresponding author.

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
