# Peer review of "Frequency and Characteristics of Non-Neurological and Neurological Stroke Mimics in the Emergency Department"

_jcm, 2023, doi:10.3390/jcm12227067_

Round 1

Reviewer 1 Report

Comments and Suggestions for Authors

The introduction lacks statistics on non-neurological and neurological stroke mimics, and also lacks information on the causes of misdiagnosis of stroke disease.

The development of neurological disorders such as aphasia and limb weakness in non-neurological patients must be explained in more detail.

In discussion, it is necessary to summarize how misdiagnosis in non-neurological and neurological stroke mimics can be reduced

Could you please explain why in Table 1, the total percentage of diagnostic brain images of non-neurological mimics was 97.6%, and the total percentage of diagnostic brain images of neurological mimics was 119.8%.

Write NIHSS without abbreviations.

Author Response

We would like to thank the reviewers for their overall positive reception and constructive comments that help to further improve the manuscript. We feel that we were able to answer all concerns and believe that the paper improved by the changes.

Comment: The introduction lacks statistics on non-neurological and neurological stroke mimics, and also lacks information on the causes of misdiagnosis of stroke disease.

Response: We appreciate the reviewer’s suggestions and have added the following passages to the introduction:

“Previous analyses stratified non-neurological and neurological stroke mimic etiologies and found that up to 44.9% of stroke mimics in patients transferred to a comprehensive stroke center for suspected stroke by Emergency Medical Services (EMS) were of non-neurological origin [8]. Demographic characteristics and clinical features such as younger age, female sex, transient or unspecific symptoms and level of consciousness alteration have found to be associated with misdiagnosis in patients with suspected stroke and it has been hypothesized that these features and thus reasons for misinterpretation may differ between subgroups of stroke mimics [9].”

Comment: The development of neurological disorders such as aphasia and limb weakness in non-neurological patients must be explained in more detail.

Response: We thank the reviewer for this important suggestion. We have explained in more detail the association between neurological symptoms in patients with primarily non-neurological disease through concomitant encephalopathy:

“Several non-neurological stroke mimic etiologies such as infections, metabolic/toxic conditions or hypertensive crisis can be associated with encephalopathy [16-19], which in turn can cause mental status alterations but also focal neurological symptoms such as speech disorders or limb paresis.”

In addition to the limitation of potential misdiagnosis, we have recognized that the rate of focal symptoms in non-neurological patients might have been overestimated due to potential misdiagnosis of actual stroke in some cases.

“In addition, patients in our emergency department might have been misdiagnosed, leading to potential over- or underestimation of the stroke mimic rate and overestimation of focal symptoms in supposedly non-neurological stroke mimics who had an actual stroke.”

Comment: In discussion, it is necessary to summarize how misdiagnosis in non-neurological and neurological stroke mimics can be reduced.

Response: We thank the reviewer for this important comment. We have added the following passage to summarize options to reduce misdiagnosis in the preclinical setting and discuss its implications for stroke care:

“Although the application of scores and use of telemedicine may refine triage protocols and bear potential of improving prehospital di-agnostic performance, our findings highlight an important caveat. Identification of stroke mimics is challenging, as etiologies and clin-ical presentations are heterogeneous and include transient events, altered mental status or focal symptoms. In patients with suspected stroke, however, a high sensitivity is considered the most important aspect when making a diagnosis, as the harm of missing a stroke outweighs the burden of preclinical overdiagnosis (false positive suspected strokes). Thus, thresholds for suspecting a stroke and transferring patients to a corresponding stroke center should remain low. The final diagnosis is up to the treating physicians of the stroke center. Furthermore, in our study more than a third of patients with non-neurological stroke mimics and nearly half with a neu-rological non-stroke etiology still had to be admitted to our hospital. These findings indicate that even beyond diagnosis and treatment of cerebrovascular events, many patients with suspected stroke still require further neurological expertise or multidisciplinary ap-proaches. Thus, transfer to comprehensive care centers is justified.”

Comment: Could you please explain why in Table 1, the total percentage of diagnostic brain images of non-neurological mimics was 97.6%, and the total percentage of diagnostic brain images of neurological mimics was 119.8%.

Response: We thank the reviewer for an opportunity to clarify this point. Patients could receive none or more than one imaging modality. Thus, percentages cannot be added up to a total and should be regarded separately for each modality. We have added the following explanatory comment to the figure legend:

As patients could receive no imaging or different imaging modalities, total percentages do not add up to 100%.

Comment: Write NIHSS without abbreviations.

Response: The abbreviation NIHSS has been written out in the running text.

Reviewer 2 Report

Comments and Suggestions for Authors

Escolà et al. conducted a study analyzing the differences between neurological and non-neurological stroke mimics among patients admitted to a comprehensive stroke center in Germany and tried to identify predictors of both the above-mentioned conditions. The manuscript reads well, but I have serious concerns about some points:

1)      The authors should extend the introduction and clarify the primary objective of the study.

2)      It is not clear what type of patients were evaluated: Those transported by emergency services to hospital? Why did patients present on their own initiative, those with suspected stroke internally, those transferred from other hospitals were excluded? These a priori exclusion without clear explanation could lead to selection biases and erroneous results. Furthermore, the authors should add “in-hospital stroke” (as mentioned in the discussion) among the exclusion criteria and adult age ( 18 years) among the inclusion criteria.

3)      Did the two stroke neurologists (J.K.E. and B.F.) who reviewed the medical records perform an inter-rater agreement? In addition, how did they decide how to classify patients when there was no agreement between the two raters?

4)      The statistical analysis of the study is too simple: Why didn’t the author perform a multivariate analysis (binary logistic regression) in order to find independent predictors of both categories (neurological vs non-neurological stroke mimics)?

5)      The most important point is that the study results in this form have very limited clinical utility in this form. How can early recognition of neurological stroke mimics from non-neurological ones help the physicians? The authors should also compare the enire mimic population with the entire stroke population in order to find predictors of stroke misdiagnosis.

6)      The discussion is cursory and the authors should compare the findings emerging from this study with those emerged from previous literature in more depth.

7)      Among the limitations of the study, the authors should add the small sample size.

Author Response

We would like to thank the reviewers for their overall positive reception and constructive comments that help to further improve the manuscript. We feel that we were able to answer all concerns and believe that the paper improved by the changes.

Comment: Escolà et al. conducted a study analyzing the differences between neurological and non-neurological stroke mimics among patients admitted to a comprehensive stroke center in Germany and tried to identify predictors of both the above-mentioned conditions. The manuscript reads well, but I have serious concerns about some points:

Response: We thank the reviewer for his/her kind words and the constructive criticism which helped to improve the manuscript.

Comment:  1)      The authors should extend the introduction and clarify the primary objective of the study.

Response: We thank the reviewer for the opportunity to clarify the objective of this study. The following passage has been added to the introduction.

“The objective of this study was to answer three questions: (1) What is the proportion of non-neurological and neurological stroke mimics among patients transferred by EMS to an urban comprehensive stroke center for suspected stroke? (2) Do preclinical characteristics and symptoms differ in patients with non-neurological and neurological stroke mimics? (3) Which variables lead to misdiagnosis by EMS in stroke mimics of non-neurological or neurological etiology?”

Comment: 2)      It is not clear what type of patients were evaluated: Those transported by emergency services to hospital? Why did patients present on their own initiative, those with suspected stroke internally, those transferred from other hospitals were excluded? These a priori exclusion without clear explanation could lead to selection biases and erroneous results. Furthermore, the authors should add “in-hospital stroke” (as mentioned in the discussion) among the exclusion criteria and adult age (≥ 18 years) among the inclusion criteria.

Response: We thank the reviewer for this important objection and the opportunity for further clarification. As stated above, our aim was to estimate the proportion of different stroke mimic etiologies among patients with suspected stroke transferred to our comprehensive stroke center via EMS and assess differences in preclinical characteristics and symptoms. As a comprehensive stroke center, most referrals from other hospital and cities of patients with “suspected stroke” are for acute treatment of actually confirmed stroke and thus, including these patient would have underestimated the rate of stroke mimics. As for in-house strokes and patients who presented on their own initiative, we would not have been able to include preclinical parameters, and documentation of these patients lacks the standardization of EMS-suspected stroke. Under the given circumstances, we feel that our inclusion and exclusion criteria provide the most reasonable generalizability for an urban German cohort. Nonetheless, we have clarified inclusion/exclusion criteria in the introduction and added potential selection bias as a limitation in the discussion.

Comment: 3)      Did the two stroke neurologists (J.K.E. and B.F.) who reviewed the medical records perform an inter-rater agreement? In addition, how did they decide how to classify patients when there was no agreement between the two raters?

Response: This important point was clarified in the methods section. In case of differing categorization, cases were discussed between the two neurologists until a consensus reading was achieved. Since the final diagnosis was primarily made by the treating physicians and the two neurologists only performed a subsequent stratification of mimic etiologies, inter-rater agreement was not performed. The following passage was added to the methods section:

“Final diagnoses at discharge as made by the treating physician were classified as cerebrovascular events (ischemic stroke, transient ischemic attack, intracerebral hemorrhage) or stroke mimics. For this analysis, we primarily focused on patients with stroke mimics. These were independently reviewed by two stroke neurologists (J.K.E., B.F.), who further stratified the etiology as non-neurological or neurological. Final categorization was then compared and in case of differing opinions, cases were discussed between the two until a consensus was achieved.”

Comment: 4)      The statistical analysis of the study is too simple: Why didn’t the author perform a multivariate analysis (binary logistic regression) in order to find independent predictors of both categories (neurological vs non-neurological stroke mimics)?

Response: As we sought to provide an exploratory characterization of non-neurological and neurological stroke mimics, we refrained from a multivariate regression analysis in our first version, considering that the nature of our study warrants much caution when interpreting such an analysis. We have now introduced uni- and multivariate multinominal logistic regression analysis to investigate predictors of non-neurological and neurological stroke mimics. The limitations section has also been extended correspondingly.

Comment: 5)      The most important point is that the study results in this form have very limited clinical utility in this form. How can early recognition of neurological stroke mimics from non-neurological ones help the physicians? The authors should also compare the enire mimic population with the entire stroke population in order to find predictors of stroke misdiagnosis.

Response: As stated above, the stroke population was included for multinominal logistic regression analysis as well as comparisons of entire mimic with entire stroke population. Discussion of the findings and inherent limitations were added to the manuscript correspondingly.

Comment: 6)      The discussion is cursory and the authors should compare the findings emerging from this study with those emerged from previous literature in more depth.

Response: Findings have been discussed and compared to previous studies in more depth, especially with regard to predictors of stroke mimics, adding (among others) the following passage to the discussion:

“Several authors have also assessed predictors of being diagnosed with a stroke mimic, among which absence of vascular risk factors, a younger age and female sex have been highlighted in particular [2,11,12]. While our study confirmed many of these previous observations, it provides closer insight towards differences in these predictors among distinct stroke mimic etiologies. While chronic kidney and liver disease (positive) as well as atrial fibrillation and hypertension (negative) were independent predictors of being diagnosed with a stroke mimic regardless of its etiology, other comorbidities (heart failure, diabetes, malignancy) as well as prehospital vital signs were exclusively associated with non-neurological stroke mimics. Increasing age, in turn, was an independent negative predictor of neurological stroke mimics only. While these findings could help clinicians to identify different types of stroke mimics in certain cases, this more differential perspective on stroke mimic predictors highlights that their application on an individual patient level warrants caution. In our study, some previously described predictors were only present in a certain subgroup of stroke mimics (age, prehospital blood pressure), while others were not at all associated with a stroke mimic diagnosis (female sex). These important barriers might become more important in the future, as health care providers are likely to face higher numbers of older and more comorbid patients with stroke mimics due to demographic changes [13].”

Comment: 7)      Among the limitations of the study, the authors should add the small sample size.

Response: We thank the reviewer for remarking this important limitation. We have added the following sentence to the discussion:

„Despite screening a large number of consultations for suspected stroke, the sample size of stroke mimic subgroups remained relatively small […].“

Round 2

Reviewer 1 Report

Comments and Suggestions for Authors

The work is of great practical and scientific interest. The topic is very relevant.

The title of the article matches the content. The purpose and objectives of the work are fully realized.

Discussion and conclusions follow logically from the results of the study and are fully consistent with the purpose of the study.

The main findings as related to the overall purpose of the study are discussed and explained in detail.

Conclusions is directly related to the data that was collected and analyzed.